# Synthesis and Physicochemical Characterization of Polymer Film-Based Anthocyanin and Starch

**DOI:** 10.3390/bios12040211

**Published:** 2022-04-01

**Authors:** Kana Husna Erna, Wen Xia Ling Felicia, Joseph Merillyn Vonnie, Kobun Rovina, Koh Wee Yin, Md Nasir Nur’Aqilah

**Affiliations:** Faculty of Food Science and Nutrition, Universiti Malaysia Sabah, Jalan UMS, Kota Kinabalu 88400, Sabah, Malaysia; mn1911017t@student.ums.edu.my (K.H.E.); felicialingling.97@gmail.com (W.X.L.F.); vonnie_merillyn_joseph_mn20@iluv.ums.edu.my (J.M.V.); weeyin@ums.edu.my (K.W.Y.); aqilah98nash@gmail.com (M.N.N.)

**Keywords:** natural dyes, anthocyanin, starch, freshness, meat quality, pH indicator

## Abstract

Colorimetric indicators, used in food intelligent packaging, have enormous promise for monitoring and detecting food quality by analyzing and interpreting the quality data of packaged food. Hence, our study developed and characterized a biopolymer film based on starch and anthocyanin for prospective meat freshness monitoring applications. The developed film was morphologically characterized using different morphology instruments to identify the interaction between anthocyanin and starch. The color differences of the proposed film in response to different pH buffers have also been investigated. The combination of anthocyanin and starch produces a smooth and homogenous surface with an intermolecular hydrogen bond that increases the biopolymer’s wavelength. The film indicated bright red at pH 2.0–6.0, bluish-grey at pH 7.0–11.0, and yellowish-green above 11.0 that the naked eye can see. The indicator film shows high sensitivity toward pH changes. The inclusion of anthocyanin increases the biopolymer film’s thickness and crystalline condition with low humidity, water solubility, and swelling values. As a result, the polymer film can be employed in the food industry as an affordable and environmentally friendly indication of meat freshness.

## 1. Introduction

Nowadays, the meat industry faces numerous challenges in preserving the quality and consistency of meat, as it spoils quickly through storage and transportation. Meat quality and safety is now a significant concern of the global food industry, as it is linked to economic and public health [1]. The freshness of meat degrades as microbial spoilage, and biochemical reactions emerge during storage [2]. High temperatures promote microbial spoilage in meat and produce proteases that degrade proteins by enzymatic decarboxylation [3,4,5]. This process produces chemical substances including ammonia, amines, total volatile basic nitrogen (TVBN), trimethylamine (TMA), and mercaptan [6,7]. They act as markers for evaluating meat freshness. Previous studies have stated that it requires specific conditions for the meat to be spoiled with micro-organisms. It was shown that the growth of micro-organisms such as bacteria, yeasts, and molds depends on the temperature [8].

The natural pigment anthocyanin is one of the most popular dyes containing high phenolic compounds that brighten the color of fruits and vegetables and undergoes structural changes at different pH environments [9,10]. Anthocyanins are usually obtained from various plants, including roselle, grape skin, red cabbage, and others [11]. The intensity and stability of anthocyanin pigments can be affected by various aspects, such as dye structure, pH, temperature, light, and other pigments such as enzymes, oxygen, sugar, and sugar metabolites. Anthocyanins are antioxidants that act as reducing agents, preventing other molecules from oxidizing [12]. Anthocyanins have more potent antioxidant activity than other flavonoids, influenced by their B-ring hydroxyl groups [6,13]. Recently, anthocyanins have been extensively used in developing a pH indicator film in monitoring the freshness of meat due to its high sensitivity response towards pH. Previously, Kang, S. et al. [14], Merz, B. et al. [15], and Alizadeh-Sani, M. et al. [16] incorporated anthocyanin onto the film matrix to improve the mechanical and barrier properties of the films, as well as to serve as a meat freshness colorimetric indicator. Additionally, the fabrication of anthocyanins improved antioxidant activity and color stability [17]. Anthocyanin is responsible for shifting color from red to blue and green when exposed to varying pH levels [18].

Cornstarch was chosen as the carrier in this work; it is the most widely used plant polysaccharide for generating edible coating films because of its availability, affordability, and excellent film-forming properties [19]. Cornstarch is a natural biopolymer commonly employed to develop environmentally friendly packaging materials. Several natural components, including plant extracts, essential oils, and other biopolymers, have been added to enhance cornstarch films’ physical and functional properties [20]. Due to the sensitivity of anthocyanin, when combined with starch film, it may be of tremendous practical benefit to the food industry as intelligent packaging, particularly in the meat industry, for measuring meat freshness [21]. However, changes in external factors such as high humidity might cause the hydrophilic nature of starch to absorb moisture, resulting in poor performance, particularly in water vapor permeability and tensile strength [22]. The film might swell and become a jelly-like structure when in contact with the surface of the meat due to moisture uptake. The thickness of a film is a key component in determining its opacity, water vapor permeability, mechanical density, and its function as a freshness indicator [23]. The thickness of a film plays a significant part in developing high-sensitivity sensor film. Sensitivity is improved with a relatively thin film.

This study synthesized and characterized a pH indicator film composed of anthocyanins and starch to monitor the freshness of meat. Different morphological instruments such as Fourier-transform infrared spectroscopy, scanning electron microscopy, transmission electron microscopy, field-emission scanning electron microscopy, and X-ray diffraction were deployed to characterize the morphological structure of the developed film. Additionally, the instruments were used to identify, structure, and investigate all the possible interactions or linkages between starch and anthocyanins. The anthocyanin extract was analyzed at various pH levels, and the indicator film’s storage stability was also investigated.

## 2. Materials and Techniques

### 2.1. Materials

Both dried roselle calyx and cornstarch were purchased from a local supplier in Sabah, Malaysia. Ammonia solution (25%, *v*/*v*), hydrochloric acid solution (37%, *w*/*v*), potassium chloride, sodium acetate, and sodium hydroxide pellets (purity ≥ 99%) were purchased from Merck Millipore (Darmstadt, Germany). Buffers with a pH range of 1.0–14.0 were formulated with sodium dihydrogen phosphate and disodium hydrogen phosphate. All chemicals employed in this study were of analytical grade. 

### 2.2. Extraction and Determination of Total Anthocyanins Content (TAC)

According to Huang, J. et al. [24], anthocyanin was extracted from ground, dried roselle calyx. A total of 15 g of the roselle powder was mixed with dH_2_O (1:10 g/mL) in a 250 mL beaker and placed in a water bath for 30 min at 50 °C with 100 rpm. The extract was filtered and stored under 4 °C in the dark for the subsequent analysis. The total anthocyanin content (TAC) was measured using the pH-differential technique [25,26], where the TAC was calculated as delphinidin 3-glucoside. Anthocyanin extracts were liquefied in a KCl buffer (0.025 M, pH 1) and C_2_H_3_NaO_2_ (0.4 M, pH 4.5) with a ratio of extract to buffer of 1:10 (*v*/*v*) for 15 min. The TAC was calculated using the formulas below:$$A\;=\;\left(A510-A700\right)\mathrm{pH}1-\left(A510-A700\right)\mathrm{pH}4.5$$
$$\mathrm{TAC}=\frac{\left(A\times \mathrm{MW}\times \mathrm{DF}\times 1000\right)}{\varepsilon \times l}$$
where,

A = diluted sample absorbance.

MW = molecular weight (delphinidin 3-glucoside: 500.83 g/mol).

DF = dilution factor (10).

ε = molar absorptivity (ε = 26,900).

l = width of cuvette (1 cm).

### 2.3. Fabrication of pH Indicator Film Incorporating of Starch

The biopolymer film was prepared using a casting method, according to Qin, Y. et al. [27], with some modification where 10 mL of roselle extract was mixed with 0.5 g of corn-starch and heated at 100 °C until gelatinized. Then, 3 mL of the film solution was pipetted into the Petri dish and left to dry at room temperature for 24 h.

### 2.4. Characterization of AS Films

#### 2.4.1. Scanning Electron Microscopy (SEM), Transmission Electron Microscopy (TEM), and Field Emission Scanning Electron Microscopy (FESEM)

The surface and cross-section of the biopolymer films were visualized using SEM, FESEM, and TEM. For SEM, the indicator film was visualized using scanning electron microscopy (S-3400N, Hitachi High-Technology, Tokyo, Japan) with an accelerating voltage of 15.0 kV. The film was coated with gold before the analysis to improve the conductivity of the sample surface. As for FESEM, the film samples were analyzed using FESEM (JSM-7900F Schottky) with ×500 magnification. The film samples were coated with palladium platinum for FESEM to enable and improve the imaging of samples. Anthocyanins/starch suspension samples were dropped onto a carbon-coated copper grid and left to dry for 12 h before being analyzed under TEM. The cross-section and particle size of the film samples were visualized using TEM (G2 Spirit Biotwin, Tecnai) with 120 kV.

#### 2.4.2. Fourier Transform Infrared (FTIR) Spectroscopy and X-ray Diffraction (XRD)

FT-IR spectra of the films were analyzed using the Agilent Technologies Cary 630 FT-IR at a spectral range from 4000 to 600 cm^−1^. The XRD of the films were operated using a SmartLab X-ray diffractometer. The film was scanned between 2θ = 3~90° with a 4.00°/min scanning rate. The film was cut into strips (1.8 mm × 1.8 mm) before the measurement.

#### 2.4.3. Thickness and Moisture Content

The thickness was determined using a micrometer at five random positions in the film to calculate the average. According to Merz, B. et al. [15], the moisture content was determined gravimetrically with some alteration in terms of size of the film samples. The moisture content was measured by weighing the samples (2.5 cm × 2.5 cm) to obtain the initial mass. The film was stored at 105 °C for 24 h and the final mass was recorded. The moisture content of the film was calculated using the following equation:$$\mathrm{MC}=\frac{\left(W-D\right)}{W}\times 100$$
where,

W = initial weight of the film.

D = final weight of the film.

#### 2.4.4. Water Solubility and Swelling Index

The indicator film’s water solubility and swelling index were determined gravimetrically by Chen, H.Z. et al. [28] and Dong, H. et al. [29], with some adjustment relating to the size of the film samples and the volume of the distilled water. The samples were cut into square film pieces (2.5 cm × 2.5 cm) and immersed in 20 mL of distilled water at room temperature for 24 h. After 24 h, the samples were taken and immediately weighed (M_0_). The remaining film was dried at 105 °C for 24 h and the final film mass measured (W_1_). The swelling index was measured by cutting the film into a circle with a diameter of 1.5 cm (SI_0_) and then immersing this in 20 mL of distilled water and leaving it at room temperature for 24 h. Then, the film (SI_1_) size was measured immediately. The water solubility and swelling index were calculated using the equations below:$$\mathrm{WS}=\frac{\left(M_{0}-W_{1}\right)}{M_{0}}\times 100$$
$$\mathrm{SI}=\frac{\left(S_{1}-S_{0}\right)}{S_{0}}\times 100$$

#### 2.4.5. Storage Stability

The stability of the film was achieved for 28 days using a gravimetric method. Five film replicates were placed in one container under three different conditions: room temperature (25 °C), incubator (37 °C), and cold room (4 °C). After weighing the initial mass on Day 0, the films were weighed on Days 7, 14, 21, and 28 to obtain the final mass.

#### 2.4.6. pH-Sensitive

The pH sensitivity of the films was determined using the Qin, Y. et al. [30] and Wu, J. et al. [31] methods with slight modification in terms of the pH level, from pH 1–14. Anthocyanin was mixed with a different pH buffer (pH 1–14), and the color spectrum was immediately measured using a UV-Vis spectrophotometer. The digital camera recorded color changes.

## 3. Results and Discussion

### 3.1. Total Anthocyanin Content (TAC)

The total anthocyanin concentration (TAC) of roselle extract was determined to be 315.76 mg/g. Roselle has a high anthocyanin concentration and an anthocyanin output of 88% [32]. Previously, Goufo, P. and Trindade, H. [33] showed that the TAC of black rice types ranged from 4.1 to 256.5 mg/g. According to the findings, roselle has amongst the highest contents of anthocyanins. Figure 1 depicts the anthocyanin absorption spectrum and the pH differential, with the maximum absorbance (max) at 520 nm. At pH 4.5, anthocyanin absorbs less light due to the hydration of the C-ring of the anthocyanin structure, and the positive charge is neutralized [34]. It will also transform to the more stable colorless carbinol when exposed to pH 4.5 and higher [35]. The anthocyanin structure is shown stable in red flavylium cation at pH below 2 [12]. The TAC can vary due to the varying color strength, types, and environmental factors [25,34].

### 3.2. Color and Spectra of the Anthocyanin at Various pH

Figure 2a depicts the color change of anthocyanin at different pH buffers, as seen with the naked eye. As the pH rises from 1.0–6.0 to 7.0–10.0 to 11.0–14.0, the anthocyanin color changes from red to bluish-gray to yellow, respectively. This occurred due to anthocyanin structure alteration and the anhydrous base structural form [17]. The occurrence of red in acidic environments is caused by the dominant form of the original anthocyanin structure flavylium cation. According to a recent study, pH is one of the most important elements determining anthocyanin coloration because the ionic nature of molecules induces reversible or permanent changes in their structure depending on the pH [36].

As the pH rises over 5, the red hue fades and turns bluish-gray due to the development of a quinoidal base [25]. Previously, Liu, B. et al. [37] reported that flavylium cations are the primary form at pH 1–3, displaying a red color, whereas deprotonation and hydration generate a carbinol pseudo base at pH 4–5 (Figure 2c). The quinoid base is formed at pH 7–8 and has a blue-purple hue [38]. As the pH rises from 12.0 to 14.0, the color changes to yellowish-green because anthocyanin degrades readily at high pH, causing the solution to turn green or yellow [21]. Anthocyanins were susceptible to pH where the anthocyanin skeleton pyrylium ring was rapidly opened and produced a chalcone structure due to their vulnerability to hydration and oxidation at the C2 position [11,39].

Figure 2b shows the absorption spectra of anthocyanin solutions altered with different pH buffers. The maximum peak was around 520 nm at pH 1.0; however, when the pH was elevated to 11.0, it was lowered to 587 nm. The absorbance dropped as the pH increased from 2.0 to 7.0 because of the high quantity of structural flavylium cation. The λmax of pH 3–5 was discovered to be about 520 nm. However, decreasing absorbance occurs, suggesting that the flavylium cation is hydrated, resulting in the formation of carbinol, which eventually achieves equilibrium with the colorless chalcone, reducing the intensity of the red hue until the solution is colorless [36,40]. When the pH was more than 6, the max peak occurred at 587 nm in the pH range of 8.0–11.0, and the corresponding absorbance increased. Furthermore, the absorbance at 599 nm dropped in the pH range of 12.0–14.0. Bathochromic phenomena occur when the maximum absorption peak shifts due to pH variations, affecting the chemical structure of anthocyanin [18,31].

### 3.3. Microstructure Evaluation of AS Film

Figure 3a,b show the indicator and starch films, while Figure 3c,d show SEM micrographs of indicator and starch films, respectively. In comparison, the anthocyanin film has a homogenous surface, indicating that the components are well spread among each other [41]. By coupling hydrogen and electrostatic connections between anthocyanin hydroxyl groups and starch, anthocyanin enhances hydrogen bonding and improves film compatibility [17,42]. This means that anthocyanin is highly compatible with starch.

Energy-dispersive X-ray spectroscopy (EDX) was used to assess the elemental composition of the indicator and the starch film [43]. Carbon and oxygen element percentages in indicator and starch films were 60.93 wt% and 23.96 wt%, and 54.54 wt% and 37.51 wt%, respectively, according to EDX analysis. Due to the substantial contacts between anthocyanin and starch, bonding between anthocyanin hydroxyl groups and starch hydroxyl groups occurred, resulting in a low number of polymer chain molecules and improved indicator film stability [21]. Calcium was found in the indicator film, whereas sulfur was found in the starch film, with 0.48 wt% and 4.29 wt%, respectively.

Figure 3e,f show FESEM micrographs of the films where the surface of the indicator film is smoother and more homogenous than starch. Due to the sheer strong molecular contact between anthocyanin and starch, the indicator film has a higher solidity than starch film [44]. The results demonstrate that indicator films have a high degree of crystallinity, validated by XRD analysis (Figure 4). Starch contains narrow pores with high compatibility, which enhances anthocyanin stability [45]. The starch film’s granular structure gave it a rougher surface, and it dissolved when mixed with anthocyanin. Incorporating anthocyanin into the starch film may improve film firmness by forming hydrogen and electrostatic bonds between anthocyanin and the starch [46].

Figure 3g–i show TEM micrographs of anthocyanin extract, starch film, and indicator film, respectively. As illustrated in Figure 3i, the result exhibits irregular shapes of phenolic rings with an average diameter of 40.5–173 nm for the indicator film. Meanwhile, most anthocyanin particles possessed a phenolic molecule ring shape, with diameters ranging from 78.9 to 163 nm (Figure 3g). Anthocyanin is an unstable pigment due to its phenolic hydroxyl groups being easily oxidized into quinones [46]. The presence of anthocyanin promotes hydrogen bonding with the starch matrix, reducing starch aggregation. Combining anthocyanin phenolic hydroxyl groups with hydroxyl groups in starch minimizes the number of chain molecules in polymers and improves their durability [42]. Anthocyanin stability can be influenced by various circumstances, including pH, enzymes, light, and oxygen [47]. In addition, Figure 3h depicts a TEM micrograph of a starch film with adherent and agglomerated particles. The aggregation of submicron-sized spherical particles with irregularly shaped particles on the wall and between themselves was seen on a wide scale [48]. When anthocyanin is added, starch granules spread and emerge in various sizes and shapes (Figure 3i), which can be linked to genetic variances and geographical differences [49].

### 3.4. XRD Analysis of Indicator Films

The XRD spectra of the film samples are depicted in Figure 4. The results demonstrate that both films’ most substantial peak, with a broad peak around 2θ, ranges from 29° to 50°, indicating semi-crystalline amorphous material behavior. They were primarily amorphous, with a weak crystalline phase [27]. According to Favaro et al. [50], the presence of a broad diffraction peak (2θ = 5–50°) in the crude anthocyanin extract XRD spectra indicates the dominance of amorphous material in the crude extract. The gelatinization process thoroughly disrupted the semi-crystalline shape of starch granules, resulting in the amorphous structure of the starch film [30]. From the results, both the indicator and starch films have hexagonal crystalline forms. The indicator film is in the same amorphous condition as the starch films but has a broader peak due to the hydrogen bond established between anthocyanin and starch [51].

### 3.5. FT-IR Analysis of Indicator Films

Figure 5 shows the FT-IR spectrum of indicator and starch films. From the results, the broad band at 3267.97 cm^−1^ in the spectrum of indicator film is attributed to O-H stretching as all materials have hydroxyl groups. Additionally, the peak at 2921.23 cm^−1^ corresponded to medium C-H stretching indicating the aromatic ring deformation [17]. The bands at 1735.59 cm^−1^ and the peak at 1718.81 cm^−1^ were attributed to the strong stretching C=O, whereas the pull of C=N was assigned to the peak at 1618.14 cm^−1^ [40]. The C=O stretching is believed to have originated from corn starch, where amylose and amylopectin enable the film to withstand effectively and maintain its structure [52]. Moreover, the vibrational mode of angular deformation of CH_3_ was observed at peak 1332.92 cm^−1^, whereas medium bending O-H was observed at peak 1221.06 cm^−1^. The indicator film stretching C-O showed strong bands at 1148.36 cm^−1^ and 1075.65 cm^−1^, which are the characteristic bands of roselle anthocyanins [24]. A peak at 991.76 cm^−1^ shows a C=C bond stretching from the anthocyanin’s pyran ring [30]. In the starch spectra, the broad band at 3236.28 cm^−1^ was attributed to the free vibration of O-H stretching (Figure 3b), whereas the peak at 697.22 cm^−1^ was attributed to the strong bending halo compound C-Br [17].

Additionally, the starch film exhibited a strong, wide band for O-H stretching at peak 3236.28 cm^−1^. The deformation of the C-H stretching is detected at wave number 1340.37 cm^−1^, whereas the peak at 1148.36 cm^−1^ indicates asymmetrical C-stretching. Carbohydrate C-O-C ring vibrations were assigned at peaks at 928.38 cm^−1^, 857.54 cm^−1^, and 756.87 cm^−1^. The incorporation of anthocyanin results in hydrogen bonds, indicating intermolecular interaction between anthocyanin and starch, increasing the wave number [30]. The intensity of these absorption peaks changed slightly after adding anthocyanins, which was attributed to changes in the chemical and physical interaction between the aromatic rings of anthocyanins and starch [53]. The band intensity at peaks 3236.28 cm^−1^, 1340.37 cm^−1^, and 988.04 cm^−1^ increased dramatically due to intermolecular hydrogen bonds between anthocyanin and starch [49]. These findings showed that roselle anthocyanins could be successfully immobilized in starch films.

### 3.6. Physical Properties of Indicator Films

Table 1 displays the moisture content, water solubility, swelling index, and thickness of the films. According to Huang, J. et al. [24], the greater the hydrophilicity of the indicator film, the stronger the interaction between films and water molecules, which causes delays or inaccuracies in assessing meat freshness. According to Table 1, the moisture content of the indicator film did not change substantially from that of the starch film (*p* < 0.05). However, because of the interaction between hydrophilic groups and water crosslinked between anthocyanin and starch, indicator films have a lower moisture content than starch films [24]. This shows that the starch film contains numerous hydroxyl groups and can absorb moisture from the surroundings. Previously, Zhai, X. et al. [54] found that the moisture content of starch/polyvinyl alcohol-roselle anthocyanin films reduced dramatically as the anthocyanin concentration increased. This shows that the interactions between starch/polyvinyl alcohol and anthocyanins can limit the availability of hydroxyl groups in starch/polyvinyl alcohol to interact with moisture.

Table 1 demonstrates no significant variation in water solubility between indicator and starch films. Both films were water soluble, but the inclusion of anthocyanin marginally improved the starch film [54]. The indicator films showed a lower water solubility compared to starch film due to the crosslinking and more significant molecular interaction between anthocyanin and starch [17,55]. Hence, anthocyanin helps to reduce the hydrophilic properties and solubility of anthocyanin in the formed film in water. Table 1 also shows a significant difference (*p* < 0.05). The swelling index of indicator film is lower than that of starch film. The swelling index of starch is determined by its ability to bind water via hydrogen bonding, but amylopectin is more likely to swell due to the weakening of starch’s intra- and inter-molecular coherence; meanwhile, amylose works as a swelling inhibitor [49,56]. The starch film has a higher swelling index than the indicator film due to the repulsion between negatively charged phosphate groups in adjacent amylopectin chains, which weakens hydrogen bonding between chains and contributes to rapid hydration and swelling [57].

Table 1 illustrates the thickness of the film samples where there was a significant difference (*p* < 0.05) between indicator and starch films. Indicator film is thicker than starch film, with thicknesses of 0.16 mm and 0.018 mm, respectively. This suggests that a higher anthocyanin concentration could result in more complex anthocyanin–starch matrices, resulting in a thicker indicator film [58]. Jiang, X. et al. [59] previously discovered how phenolic hydroxyl groups in roselle anthocyanin interact with hydroxyl groups in PVA, decreasing the polymer’s chain molecules and increasing mechanical characteristics. Anthocyanin inclusion into the starch film via hydrogen interactions between anthocyanin and starch results in a significant interfacial contact [18,46].

### 3.7. Storage Stability

The storage stability of film samples is shown in Figure 6. The films were stored over 28 days in various spots and temperatures. The results indicate that the indicator film degrades slowly and is more stable in a cold room (4 °C) compared to room temperature and in an incubator (37 °C). The loss of biologically active anthocyanins can be slowed down by storing them at lower temperatures. The starch film showed high stability in all environments and temperatures compared to the indicator film. Anthocyanin is temperature-sensitive and temperature-dependent and rapidly degrades at high temperatures [60]. The effects of heating manifest themselves in two stages. The anthocyanin glycoside linkages are first hydrolyzed, making the aglycone unstable, and then the aglycone rings are opened, generating carbinol and chalcone groups [61]. Due to gradual water gain in the films during storage, it eventually plasticizes the polymer matrix to a greater degree. The starch film is more durable and steadily degraded, resulting in an increased chain mobility and more extensible films [62]. The indicator film becomes unstable when exposed to high temperatures for a more extended period, resulting in decreased anthocyanin concentrations but increasing polymer color accumulation, as temperature and water activity significantly impact anthocyanin degradation rates [63].

## 4. Conclusions

This study successfully developed and characterized a biopolymer film based on roselle anthocyanin and cornstarch. The structural characterization of the developed film revealed that anthocyanin and starch were miscible and could be aggregated via hydrogen bonds. Fabrication of anthocyanin onto the starch matrix greatly improved the mechanical stability and hydrophobicity of the film. The indicator film generated a sensitive response towards different pH levels as the color changed dramatically from red to bluish-gray, and then yellow when the pH increased from 1 to 14. In future, starch/anthocyanin films might be extensively used as active and intelligent sensors in the meat industry to monitor the freshness of meat such as chicken, shrimp, and pork in a timely manner.

## Figures and Tables

**Figure 1 biosensors-12-00211-f001:**
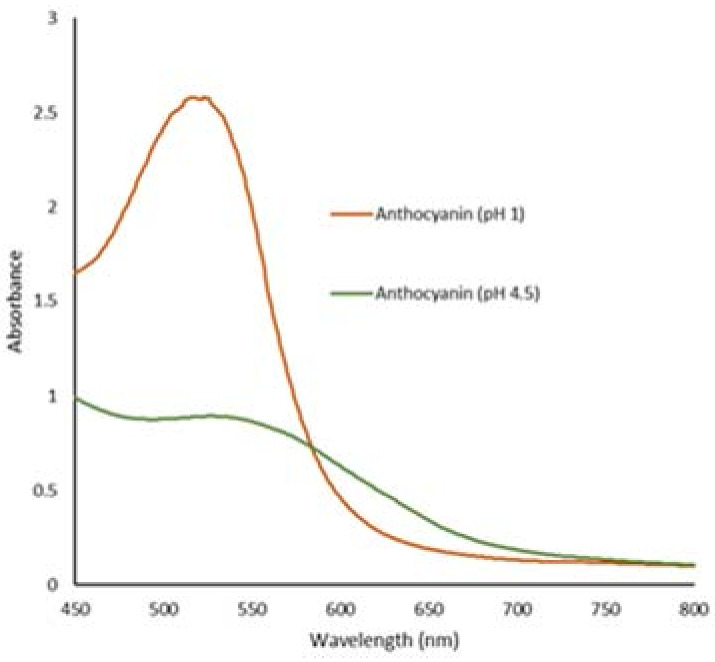
UV-Vis spectra of pH-differential of anthocyanin at pH 1 and pH 4.5.

**Figure 2 biosensors-12-00211-f002:**
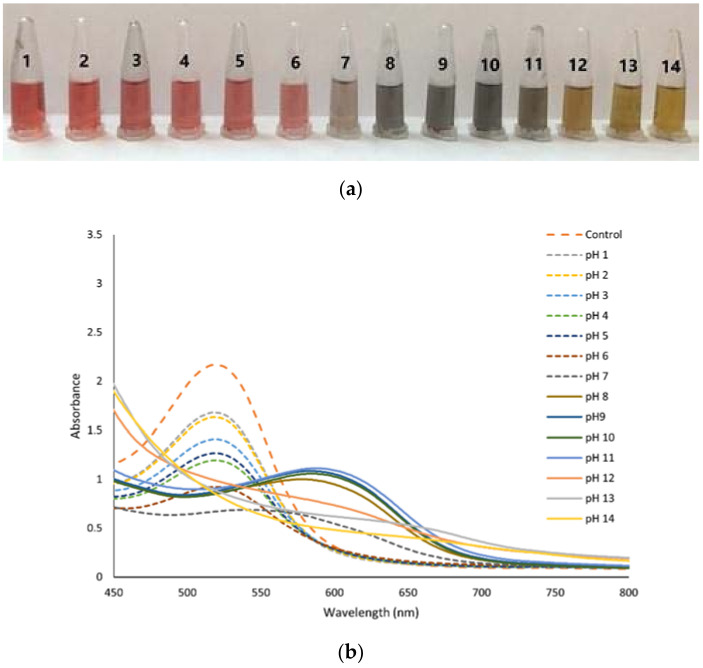
(**a**) UV-Vis spectra of anthocyanin extract in different pH buffers, (**b**) color variation of anthocyanin in different pH, and (**c**) structural transformation of anthocyanin in pH.

**Figure 3 biosensors-12-00211-f003:**
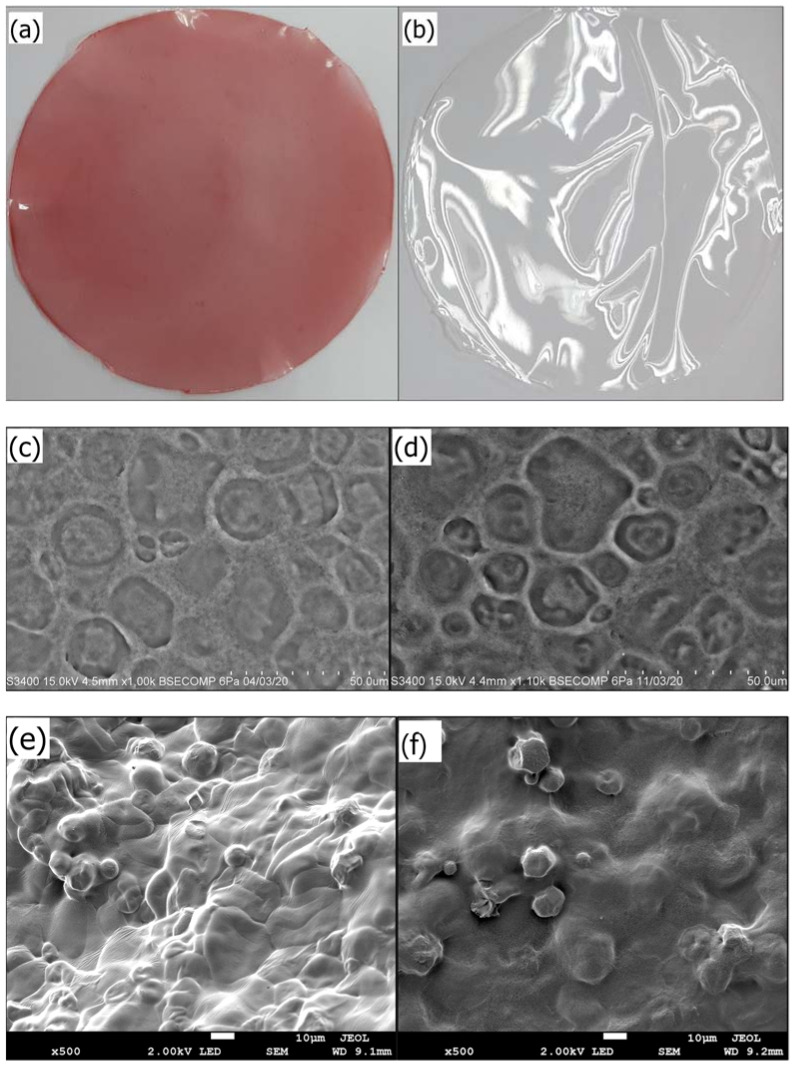
Pictures of (**a**) indicator film, (**b**) starch film; SEM micrograph of (**c**) indicator film, (**d**) starch film; FESEM micrograph of (**e**) indicator film, (**f**) starch film; and TEM micrograph of (**g**) anthocyanin extract, (**h**) starch film, and (**i**) indicator film.

**Figure 4 biosensors-12-00211-f004:**
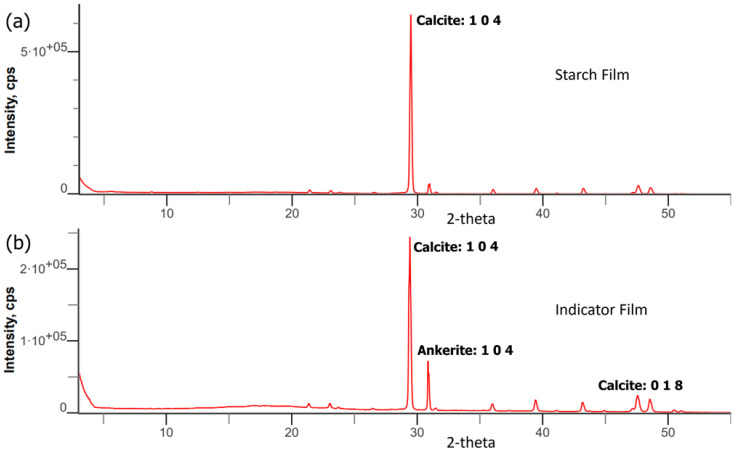
XRD spectra of (**a**) starch film and (**b**) indicator film.

**Figure 5 biosensors-12-00211-f005:**
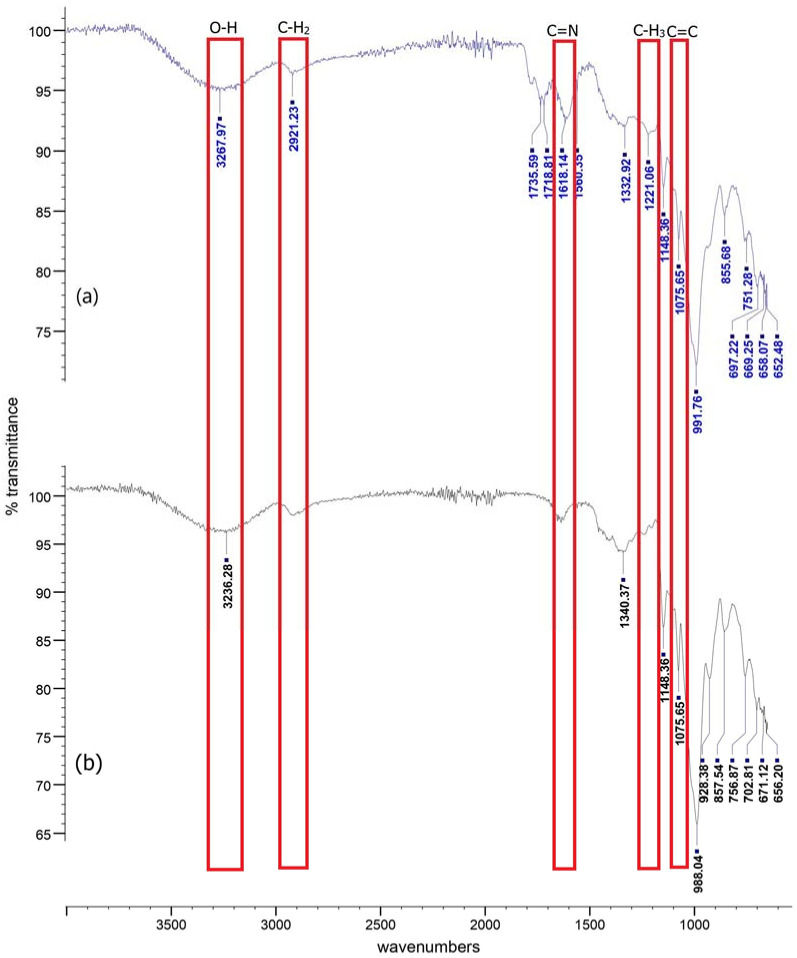
FT-IR spectra of developed films, (**a**) indicator film, (**b**) starch films.

**Figure 6 biosensors-12-00211-f006:**
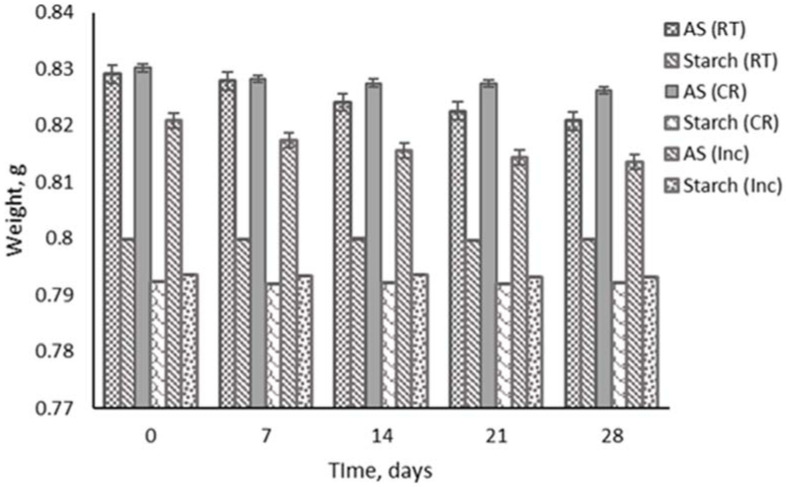
Storage stability of indicator and starch films at room temperature, in a cold room, and in an incubator for 28 days.

**Table 1 biosensors-12-00211-t001:** Moisture content (MC), water solubility (WS), swelling index (SI), and thickness (T) of indicator and starch films.

	Indicator	Starch
MC (%)	11.16 ± 11.14	13.68 ± 8.43
WS (%)	55.17 ± 1.26	55.49 ± 1.63
SI (%)	6.48 ± 0.52	14.54 ± 2.57
T (mm)	0.16 ± 0.07	0.018 ± 0.02

## Data Availability

Not applicable.

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
