# Peer review of "Synthesis and Physicochemical Characterization of Polymer Film-Based Anthocyanin and Starch"

_biosensors, 2022, doi:10.3390/bios12040211_

Round 1
Reviewer 1 Report
Using biopolymer film based on starch and anthocyanin for prospective meat freshness monitoring applications is an innovative approach.
The indicated values are average, not instantaneous values, which is of key importance for the correct detection, and of great importance for the film's stability.
Nothing was mentioned about the porosity of the films. Can changes in external factors affect the indicative nature of the material used (e.g. external humidity)?
Does the layer thickness change during application or does it stretch? Does it influence the indications?
has there been any research on this subject?
The XRD shows an indescribable signal at around 49 degrees. Are the authors able to explain from what it comes from?
Author Response
Reviewer #1:
Using biopolymer film based on starch and anthocyanin for prospective meat freshness monitoring applications is an innovative approach.
- The indicated values are average, not instantaneous values, which is of key importance for the correct detection, and of great importance for the film's stability.
- Thank you for the constructive comments. The manuscript has been revised accordingly based on the suggestion given.
- Nothing was mentioned about the porosity of the films. Can changes in external factors affect the indicative nature of the material used (e.g. external humidity)?
- This research did not do porosity analysis of the developed films. “Changes in external factors such as high humidity might cause the hydrophilic nature of starch to absorb moisture, resulting in poor performance, particularly in water vapour permeability and tensile strength (Thakur et al., 2016; Singha & Kapoor, 2014).” The detail can be found in Section 1 paragraph 3.
- Does the layer thickness change during application or does it stretch? Does it influence the indications?
- “The film might swell and become a jelly-like structure when in contact with the surface of the meat due to moisture uptake. Thickness of film is a key component determining the opacity, water vapor permeability, mechanical, density and its function as a freshness indicator (Mary et al., 2020; Oluwasina et al., 2019). Thickness of a film plays a significant part in the development of high sensitivity sensor film. Sensitivity is improved with a relatively thin film.” The detail can be found in Section 1 paragraph 3.
- Has there been any research on this subject?
- Yes, much research has been done on investigating the function of anthocyanin as a meat freshness indicator. The details can be found in Section 1 paragraph 2.
“Anthocyanins have been extensively used in developing a pH indicator film in monitoring freshness of meat due to its high sensitivity towards pH. Previously, Kang et al. (2020), Merz et al. (2020) and Wu et al. (2019) incorporated anthocyanin onto the film matrix to improve the mechanical and barrier properties of the films as well as serve as meat freshness colorimetric indicator. Additionally, fabrication of anthocyanins improved the antioxidant activity and color stability (Zhang et al., 2019). Anthocyanin is responsible for shifting color from red to blue and green when exposed to varied pH levels (Jiang et al., 2020). “
- The XRD shows an indescribable signal at around 49 degrees. Are the authors able to explain from what it comes from?
“According to Favaro et al. (2018), the presence of a broad diffraction peak (2θ = 5-50°) in the crude anthocyanin extract XRD spectra indicates the dominance of amorphous material in the crude extract.”

Reviewer 2 Report
The current paper reports the use of Anthocyanin and Starch film for food freshness monitoring system. The introduction section lacks proper background of the current study, the experimental section lacks adequate experimental details. The result/discussion section needs to be revised with proper conclusions. Based on my assessment, I suggest major revision of this paper.
The other comments are as follows:
- The title needs to be revisit. It can shorten and focused. ‘for Potential Monitoring Meat Freshness Applications’- this portion can easily be removed.
- The abstract must be checked for English and proper punctuation. For example, the first sentence should be like that: Colorimetric indicators, used in food intelligent packaging, have enormous promise for and so on.
- Line 15-17 can be omitted in abstract section. Abstract should be written in a focused way. In abstract, there is no need for sample/experimental details. It will contain the essence of the current work.
- Line 61-62 needs to be rewritten. It is hard to follow in current form.
- Line 62-63 is wrong!
- Needs additional background on anthocyanin, as film, based in literature.
- Line 69-71: concentration and purity are stated wrongly. Have a look in recent literature to make the corrections.
- Line 77: What does ‘anthocyanin extract was done’ means?
- Section 2.3: Need details of film preparation including the modification.
- In all the experimental sections, the authors have used the term ‘slight modification’ which is not scientific. Exact modification of the experimental technique must be given in details.
- Line 218: Fig. should appear and explained in chronological orders.
- Details of TEM sample preparation should be included in the revised manuscript.
- 3: All figs need scale bar. The scale bar is not readable in SEM and TEM images.
- The XRD spectrum seems somewhat abnormal (more like a hand-drawn!!). Verify the spectra and present in proper way (with both axis) with peak labelling.
- I am missing the discussion section on the paper!
- The conclusion section rather seems like a (part of) discussion section! In that case proper conclusion section is required.
Author Response
Reviewer #2:
The current paper reports the use of Anthocyanin and Starch film for food freshness monitoring systems. The introduction section lacks proper background of the current study, the experimental section lacks adequate experimental details. The result/discussion section needs to be revised with proper conclusions. Based on my assessment, I suggest major revision of this paper. The other comments are as follows:
- The title needs to be revisit. It can shorten and focused. ‘for Potential Monitoring Meat Freshness Applications’- this portion can easily be removed.
- The title has been revised and shortened to “Synthesis and Physicochemical Characterization of Polymer Film-Based Anthocyanin and Starch”
- The abstract must be checked for English and proper punctuation. For example, the first sentence should be like that: Colorimetric indicators, used in food intelligent packaging, have enormous promise for and so on.
- The abstract has been checked and revised for proper use of English and punctuation.
- Line 15-17 can be omitted in abstract section. Abstract should be written in a focused way. In abstract, there is no need for sample/experimental details. It will contain the essence of the current work.
- The sentence has been omitted in the abstract section as suggested. In exchange, new sentence has been added in the abstract section:
“The developed film was morphologically characterized using different morphology instruments to identify the interaction between anthocyanin and starch.”
- Line 61-62 needs to be rewritten. It is hard to follow in current form.
- The line in 61-62 has be re-wrote to “This study synthesized and characterized a pH indicator film composed of anthocyanins and starch to monitor the freshness of meat”
- Line 62-63 is wrong!
- The line 62-63 has been revised as stated:
“Different morphological instruments such as Fourier-transform infrared spectroscopy, scanning electron microscopy, transmission electron microscopy, field-emission scanning electron microscopy, and x-ray diffraction were deployed to characterize the morphological structure of the developed film. Besides, the instruments were used to identify, structure, and investigate all the possible interactions or linkages between starch and anthocyanins.”
- Needs additional background on anthocyanin, as film, based in literature.
- Additional background on anthocyanin have been included in Section 1 (Introduction) Paragraph 2. Added information are as follow:
“Anthocyanins have been extensively used in developing a pH indicator film in monitoring freshness of meat due to its high sensitivity towards pH. Previously, Kang et al. (2020), Merz et al. (2020) and Wu et al. (2019) incorporated anthocyanin onto the film matrix to improve the mechanical and barrier properties of the films as well as serve as seafood freshness colorimetric indicator. Additionally, fabrication of anthocyanins improved the antioxidant activity and color stability (Zhang et al., 2019). Anthocyanin is responsible for shifting color from red to blue and green when exposed to varied pH levels (Jiang et al., 2020).”
- Line 69-71: concentration and purity are stated wrongly. Have a look in recent literature to make the corrections.
- The sentence has been revised as follow:
“Ammonia solution (25%, v/v), hydrochloric acid solution (37%, w/v), potassium chloride, sodium acetate, sodium hydroxide pellet (purity ≥ 99%) were purchased from Merck Millipore (Germany).”
- Line 77: What does ‘anthocyanin extract was done’ means?
- The sentence has been revised to:
“According to Huang et al. [20], anthocyanin was extracted from ground dried roselle calyx.”
- Section 2.3: Need details of film preparation including the modification.
- The sentence has been revised accordingly.
“The preparation of the biopolymer film was done using a casting method, according to Qin et al. [23], with some modification, where 10 ml of roselle extract was mixed with 0.5 g of corn-starch and heated at 100 °C until gelatinized.”
- In all the experimental sections, the authors have used the term ‘slight modification’ which is not scientific. Exact modification of the experimental technique must be given in details.
- The exact modification of the experimental techniques have been updated and well-explained in Section 2.3 and Section 2.4.5.
- Line 218: Fig. should appear and be explained in chronological orders.
- Figure has been explained in chronological order as suggested. The amended explanation can be found in Section 3.3 and can be referred to Figure 3.
- Details of TEM sample preparation should be included in the revised manuscript.
- The details of the TEM sample preparation has been added and revised in the manuscript.
“Samples of anthocyanins/starch suspension were dropped onto a carbon-coated copper grid and let dried for 12 hours before being analyzed under TEM. The cross-section and particle size of the film sample were visualized using TEM (G2 Spirit Biotwin, Tecnai) with 120 kV.”
- 3: All figs need scale bar. The scale bar is not readable in SEM and TEM images.
- All figures have been updated with scale bars clearly shown on the images.
- The XRD spectrum seems somewhat abnormal (more like a hand-drawn!!). Verify the spectra and present in proper way (with both axis) with peak labelling.
- XRD spectrum (Figure 4) has been revised with proper labelling of x- and y-axis as well as peak labelling.
- I am missing the discussion section on the paper!
- Subtopic 3 has been revised to Results and Discussion.
- The conclusion section rather seems like a (part of) discussion section! In that case proper conclusion section is required.
- Conclusion section has been revised accordingly as follow:
“This study successfully developed and characterized a biopolymer film based on roselle anthocyanin and cornstarch. The structural characterization of the developed film revealed that anthocyanin and starch were miscible and could be aggregated via hydrogen bonds. Fabrication of anthocyanin onto the starch matric greatly improved the mechanical, stability and hydrophobicity of the film. The indicator film generated a sensitive response towards different pH levels as the color changed dramatically from red to bluish-gray, then yellow when the pH increased from 1 to 14. In future, starch/anthocyanins films might be extensively used as active and intelligent sensors in meat industry to monitor the freshness of meat such as chicken, shrimp and pork in a swift manner of time.”

Round 2
Reviewer 2 Report
- Fig. 3a and 3b, still missing the scale bar.
- Fig. 4: XRD peaks still not labelled!
Author Response
Dear Reviewer,
Thank you for the constructive comment. The authors has revise the manuscript accordingly.
- Fig. 3a and 3b, still missing the scale bar.
- Thank you for the suggestion. Scale bars of Figure 3a and 3b have been added.
- Fig. 4: XRD peaks still not labelled!
- Thank you for the reminder. The XRD peaks have been properly labelled as shown in Figure 4.